# Physiological and Molecular Characteristics of Southern Leaf Blight Resistance in Sweet Corn Inbred Lines

**DOI:** 10.3390/ijms231810236

**Published:** 2022-09-06

**Authors:** Caiyun Xiong, Haiwei Mo, Jingsheng Fan, Wenchuang Ren, Hu Pei, Yahui Zhang, Ziwei Ma, Wenyi Wang, Jun Huang

**Affiliations:** 1Guangdong Provincial Key Laboratory of Plant Molecular Breeding, College of Agriculture, South China Agricultural University, Guangzhou 510642, China; 2Guangxi Subtropical Crops Research Institute, Nanning 530001, China

**Keywords:** southern leaf blight, sweet corn, transcriptome, physiological response, molecular mechanisms

## Abstract

Southern corn leaf blight is one of the most widespread foliar diseases in maize-producing areas worldwide and can seriously reduce the yield and quality of sweet corn. However, the molecular mechanisms underlying the disease in sweet corn have not been widely reported. In this study, two sweet corn inbred lines, resistant K13 (RK13) and susceptible K39 (SK39), were used to explore the disease resistance mechanism of southern leaf blight. We observed morphological characteristics and assessed the changes in protective enzymatic activity in sweet corn leaves after inoculation of *C. heterostrophus*. RNA-seq was performed to elucidate the transcriptional dynamics and reveal the key pathways involved in southern leaf blight resistance without pathogens (Mock) and at 1 and 3 days post inoculation (1 and 3 dpi). Differentially expressed genes (DEGs) were identified in the SK39 group (including three pairwise combinations: SK39−0d_vs_SK39−1d, SK39−1d_vs_SK39−3d and SK39−1d_vs_SK39−3d), the RK13 group (including three pairwise combinations: RK13−0d_vs_RK13−1d, RK13−1d_vs_RK13−3d and RK13−1d_vs_RK13−3d), and the SK39_vs_RK13 group (including three pairwise combinations: SK39−0d_vs_RK13−0d, SK39−1d_vs_RK13−1d, and SK39−3d_vs_RK13−3d). In our study, 9455 DEGs from the RK13 group, 9626 from the SK39 group, and 9051 DEGs from the SK39_vs_RK13 group were obtained. Furthermore, 2775, 163, and 185 DEGs were co-expressed at SK39_vs_RK13, RK13, and SK39, respectively. A functional analysis of the DEGs revealed that five pathways—i.e., photosynthesis, plant hormone signal transduction, MAPK signaling pathway, phenylpropanoid biosynthesis, and biosynthesis of secondary metabolites—and transcription factor families play crucial roles in disease resistance. The results from the present study enabled the identification of the JA and SA signaling pathways, which are potentially involved in the response to southern leaf blight in maize. Our findings also highlight the significance of ZIM transcription factors and pathogenesis-related (PR) genes during pathogen infection. This study preliminarily explored the molecular mechanisms of the interaction between sweet corn and *C. heterostrophus* and provides a reference for identifying southern leaf blight resistance genes in the future.

## 1. Introduction

Southern corn leaf blight (SCLB) is a serious fungal foliar disease in maize, caused by *Cochliobolus heterostrophus* (amorphic *Bipolaria maydis*), which is prevalent in maize-producing areas worldwide [1]. Understanding the infestation process of SCLB is essential to study its resistance. The spores of *C. heterostrophus* germinate 1 h after inoculation, aerial hyphae appear 6~10 h after inoculation, and sturdy hyphae along the leaf veins appear 2 h later, indicating successful pathogen colonization. After 24~36 h of successful colonization, the mycelium expands into the cell wall space or directly invades the mesophyll cells; after 60~72 h of infection, significantly expanded lesions appear on the leaves [2,3]. The lesions of resistant varieties appear as independent, yellow-brown necrotic spots that do not intersect with each other. The lesions of susceptible varieties are often brown and oval in shape, with a large area, and the lesions spread [4]. The lesions on maize leaves directly affect photosynthesis, thereby affecting grain filling. Generally, pathogens cause a 15~20% reduction in production, and in severe cases, can lead to a production loss of more than 40% [5,6].

After pathogens infect the host, they release toxins into the body to destroy the plant cell structure. Elevated levels of reactive oxygen species (ROS) occur in response to the infection. Superoxide dismutase (SOD) and peroxidase (POD) have a certain synergy to jointly fight invading pathogens [7]. The former catalyzes the conversion of ROS into O_2_ and H_2_O_2_ via a disproportionation reaction [8], while the latter converts the generated H_2_O_2_ into H_2_O and O_2_ [9], protecting plants from damage. Polyphenol oxidase (PPO) has obvious physiological significance in explaining the rapid production of quinones due to injury, which further causes browning and death [10]. Phenylalanine ammonia lyase (PAL) has been demonstrated to be a key enzyme in the synthesis of several defense-related secondary compounds such as phenols and lignin [11]. Although plant protective enzymes can reduce the ROS damage caused by pathogens, they cannot prevent irreversible damage to plants [12].

Harnessing the genetic resistance of host plants is one of the most effective ways of controlling crop diseases. Marker-assisted selection is still an important method for mining plant disease resistance genes [13] and is widely used in the mapping of disease resistance genes of northern corn leaf blight (NCLB) [14] and southern corn leaf blight (SCLB). QTLs for resistance to SCLB have also been reported recently. A total of 11 QTLs for SCLB resistance were identified in the recombinant inbred lines constructed with B73 and Mo17 as parents, of which seven were derived from the disease-resistant parent Mo17 [15]. A total of 6 QTLs were identified using 192 inbred RIL populations to identify resistance to SCLB at the seedling stage [16]. Four shared QTLs were identified from the B73 × Mo17 recombinant inbred line population in the four environments, of which two were located in the common region bin3.04 and the other two were located in bin1.10 and bin8.02/03 [17]. The SCLB resistant line NC250P and susceptible line B73 were used to construct a near-isogenic line population. NC292 and NC330 lines carrying the B73 background showed that introgression of three gene segments (bin3.03/04, bin6.01, and bin9.02/03) significantly increased resistance to SCLB [18]. Currently, a single recessive resistance gene against race O was identified in a Nigerian maize inbred line, named *rhm1* [19]; accordingly, *rhm1* was transformed into various maize inbred lines as a source of resistance to SCLB [20]. It was reported that *rhm1* was located on the short arm of chromosome 6 between the AFLP marker p7m36 and the RFLP marker agrP144, and the genetic distances between rhm1 and p7m36 and agp144 were 1.0 cM and 0.5 cM, respectively [21]. The RFLP probe UMC85 was linked to rhm1 with a genetic distance of 1.5 cM [22]. However, the genetic basis and disease resistance mechanisms of SCLB are still unclear.

Recently, the development of transcriptome has facilitated the study of disease resistance mechanisms. Transcriptome data of tobacco varieties, 4411-3 (highly resistant) and K326 (moderately resistant) inoculated *R. solanacearum* at 0, 10, and 17 days showed that defense-related and phytohormone-related genes generally showed genotype-specific regulation and expression differences in tobacco response to *R. solanacearum* [23]. An integrated mRNA and microRNA transcriptome analysis revealed that the transduction of pathogen signals, transcriptional reprogramming, induction of hormone signaling, and activation of pathogenesis-related (PR) genes were involved in response to PVA in potato [24]. Transcriptome studies of susceptible and resistant rice varieties against blast fungus indicated that rapid and high-amplitude transcriptional reprogramming was essential for *M. oryzae* defense, and that the protein translation machinery was regulated at an early stage in rice immune responds [25]. Although interaction transcriptome dynamics between plants and pathogens have been reported, to our knowledge, no transcriptome studies have focused specifically on how sweet corn responds to SCLB.

In this study, we observed the morphological characteristics of, and assessed the changes in, the protective enzymatic activity (SOD, POD, PPO, PAL) of sweet corn leaves after inoculation with *C. heterostrophus*. We also conducted time-series transcriptomes to investigate SCLB resistance in the susceptible (SK39) and the resistant (RK13) sweet corn lines. These results provide insights into the SCLB response mechanism in sweet corn during pathogen infection.

## 2. Results

### 2.1. Morphological Characteristics and Changes in Protective Enzymatic Activity of Sweet Corn Leaves after Inoculation of C. heterostrophus

After inoculation with the pathogen, the leaves of both lines developed typical the symptoms of corn blight (Figure 1A,B). Punctuate lesions appeared on the leaves of the resistant line RK13 at 1 dpi, while larger spots emerged in the middle of the leaves of the susceptible line SK39. Over time, we observed yellowing of the RK13 leaves and the appearance of more and larger spots at 3 dpi. However, the lesions in SK39 showed a spreading trend, and the tips of the leaves appeared dry and necrotic. We measured changes in several protective enzymatic activities (Figure 1C–F) and the gene expression patterns of these enzymes were determined (Appendix A). The results showed that the activities of SOD and POD both increased after inoculation with pathogens, among which SOD increased first and then decreased (Figure 1C), whereas the activity of POD increased with pathogen infection time (Figure 1D). Although the activities of PPO and PAL were decreased after inoculation with the pathogens, this was not associated with cultivar resistance (Figure 1E,F).

### 2.2. Overview of mRNA Sequencing Data

To elucidate the response mechanisms of resistance to SCLB in sweet corn, we generated 18 mRNA sequencing libraries from three biological replicates for leaves without pathogens (Mock) and at 1 and 3 dpi, respectively, and sequenced them on an Illumina HiSeq™ 2500 platform. The detailed sequencing data are shown in Table 1. After quality control, clean read form samples ranged from 44.84 M to 46.24 M, error rates were below 0.025%, and Q30 (%) were both higher than 94.37%. After mapping these clean reads to the B73 reference genome (version 4), the “multiple mapped” reads ranged from 2.93% to 3.73%, and the “unique mapped” reads ranged from 78.06% to 86.89%. These results indicated that the quality of the mRNA sequencing data was high and could be used for subsequent analysis.

### 2.3. Identification of Differentially Expressed Genes (DEGs)

To visualize the variations and similarities among the samples, we performed a principal component analysis (PCA) (Figure 2A) on the normalized FPKM values of all detected genes. The results showed that the three biological replicates clustered closely and were separated by time point, treatment, and genotype. To investigate the expression profiles of genes in maize leaves in response to *C. heterostrophus* infection, expression levels were compared based on the FPKM values. Using DESeq2 software, the gene expression between the samples (1 dpi and 3 dpi) and Mock (without pathogens) were compared. DEGs were identified using nine pairwise combinations of three groups (SK39, RK13, and SK39_vs_RK13) (Figure 2B). The first two groups consisted of three pairwise combinations (SK39/RK13−0d_vs_SK39/RK13−1d, SK39/RK13−1d_vs_SK39/RK13−3d, and SK39/RK13−1d_vs_SK39/RK13−3d), while the SK39_vs_RK13 group was composed of SK39−0d_vs_RK13−0d, SK39−1d_vs_RK13−1d, and SK39−3d_vs_RK13−3d.

As shown in Figure 2B, whether it was SK39 or RK13, the expression levels of numerous genes changed after *C. heterostrophus* infection (SK39 had 8032 genes at 1 dpi and 7972 genes at 3 dpi; RK13 had 7723 genes at 1 dpi and 8129 genes at 3 dpi). Although an increase in the number of genes was observed from 1 to 3 dpi, the change was not significant (351 DEGs in RK13, 468 DEGs in SK39). At this stage, genes were up-regulated in both SK39 and RK13 (both 254 genes), particularly in the latter. From the SK39_vs_RK13 group, we found that more genes participated in the disease resistance process at 3 dpi than 1 dpi (5725_vs_4772). Overlapping the DEGs between the samples and WT showed that 2575, 163, and 185 DEGs were co-expressed at SK39_vs_RK13, RK13, and SK39, respectively (Figure 2C–E).

### 2.4. Functional Classification of Differentially Expressed Genes (DEGs)

Plant disease resistance is a complex biological process. In our study, 9455 DEGs from the RK13 group, 9626 from the SK39 group, and 9051 from the SK39_vs_RK13 group were obtained (Figure 2C–E). To further study the biological roles of DEGs in resistance to southern corn leaf blight, we conducted GO analyses of the DEGs identified in this study. The Top 30 GO terms and Top 30 KEGG pathways are listed in Appendix A, respectively, and heatmaps show terms and pathways that were enriched in two or more comparison groups (Figure 3).

The GO enrichment results of the RK13 and SK39 groups were highly similar, whereas in the SK39_vs_RK13 group, the significantly enriched terms in both groups were not significant (Figure 3A). “Cation binding”, “iron ion binding”, and “ion binding” were the three most significant terms in the SK39_vs_RK13 group, whereas “cation binding” in the RK13 group and the two terms “iron ion binding” and “ion binding” in the SK39 group were more significant than those in the SK39_vs_RK13 group.

Ten pathways were more enriched in the SK39 and RK13 groups (Figure 3B), i.e., “MAPK signaling pathway”, “circadian rhythm”, “ribosome” and five photosynthesis-related pathways (photosynthesis, photosynthesis-antenna proteins, carotenoid biosynthesis, carbon metabolism, and carbon fixation in photosynthetic organisms), and two amino acid metabolism pathways (alanine, aspartate and glutamate metabolism, and cysteine and methionine metabolism). Two pathways, “fructose and mannose metabolism” and “cyanoamino acid metabolism”, were more enriched in the SK39_vs_RK13 and RK13 groups. Additionally, four pathways were more enriched in the SK39 and the SK39_vs_RK13 groups, namely, “ubiquinone and other terpenoid-quinone biosynthesis”, “alpha-linolenic acid metabolism”, “stilbenoid, diarylheptanoid and gingerol biosynthesis”, and “limonene and pinene degradation”.

### 2.5. Clustering Analysis of the Co-Expressed DEGs

After filtering out duplicated genes, 2499 co-expressed DEGs were obtained from the three groups (SK39_vs_RK13, RK13, and SK39). To explore the expression patterns of these DEGs, we performed a temporal analysis of the DEGs of each variety using Short Time-series Expression Miner (STEM) software with the default parameters. DEGs were classified according to expression trends, and 15 trends were identified (Appendix A). Three significant trends (*p* < 0.05) contained 2108 DEGs, accounting for more than 84% of the total (Figure 4A). Profiles 0 (863 DEGs, 34.5%) and 14 (984 DEGs, 39.4%) represented two typical trends, that is, with an increase in infection time, negatively regulated genes constantly decreased in expression, whereas positive regulators acted by increasing their expression levels. Profile 7 (261 DEGs, 10.4%) represents a special class of genes expressed at low levels in the susceptible cultivar which responded only in the initial stages of pathogen invasion.

To evaluate the potential functions of these co-expressed DEGs in the response of sweet corn to SCLB, GO annotation and KEGG pathway analysis were performed. The GO results showed that two photosynthesis-related terms, “photosynthesis, light harvesting” and “photosynthesis, light reaction”, were significantly enriched (Figure 4B). Notably, “defense response”, “response to biotic stimulus” and “gene silencing” were also significantly enriched; therefore, we speculated that these two terms may be related to Profiles 0 and 14, enhancing resistance from direct activation of resistance genes and inhibiting susceptible genes, respectively.

The KEGG results also showed that two photosynthesis-related pathways, “photosynthesis-antenna proteins” and “carbon fixation in photosynthetic organisms”, and the “peroxisome” were significantly enriched (Figure 4C). This result was consistent with the GO results, indicating that photosynthesis-related genes were closely related to SCLB. “Metabolic pathways”, and “biosynthesis of secondary metabolites” were the entries with the largest number of genes, with 119 and 68 genes, respectively (Appendix A). In addition, two pathways, i.e., “phenylpropanoid biosynthesis” and “plant hormone signal transduction”, have been reported to be associated with plant disease resistance [26].

### 2.6. Analysis of Important Pathways in Response to SCLB in Sweet Corn

A functional analysis of these DEGs (Figure 3) and co-expressed DEGs (Figure 4) revealed five important pathways in response to SCLB in sweet corn, namely, “photosynthesis”, “plant hormone signal transduction”, “MAPK signaling pathway”, “phenylpropanoid biosynthesis” and “biosynthesis of secondary metabolites”. We provide a schematic diagram of these pathways (Figure 5) and a schema to demonstrate how these pathways respond to SCLB (Figure 5A) and the gene expression patterns of several elements (Figure 5B). It was found that the expression levels of CaM4 and PR1 protein genes were greatly increased after inoculation with pathogens. The expression of JAZ genes changed more apparently in the resistant line RK13, especially at 1 dpi, while the four genes, *Zm00001d027900*, *Zm00001d005726*, *Zm00001d048268*, and *Zm00001d020614* also changed markedly in the susceptible line SK39 after inoculation with the pathogen.

### 2.7. Expression of Transcription Factors (TFs) in Response to SCLB

Transcription factors (TFs), as key components of transcriptional regulatory mechanisms, are involved in the initiation, regulation, and transcription of genes and may play important roles in plant growth and development. To better understand the role of TFs in response to SCLB in sweet corn, we identified 230 differentially expressed TFs from the 2499 DEGs co-expressed in the three groups (SK39_vs_RK13, RK13 and SK39) (Appendix A). The most abundant TFs belonged to the bHLH (12 genes), MYB (9 genes), NAC (8 genes), WRKY (7 genes), and ZIM (6 genes) families. Additionally, ethylene-responsive TFs (AP2-EREBP family) and auxin-responsive TFs (ARF and Aux/IAA families) were also identified. The expression patterns of TFs were investigated, and it was found that the bHLH and ZIM TF families and auxin-responsive TFs were mainly highly expressed in the resistant line RK13, while the MYB, NAC, and WRKY TF families and ethylene-responsive TFs were mainly highly expressed in the susceptible line SK39 (Figure 6). The expression level of *Zm00001d014995* was very low when inoculated with the pathogen, while invasion of the pathogen significantly increased its transcription level in SK39 (Figure 6A). One gene, *Zm00001d036551*, was highly expressed before RK13 was inoculated with the pathogen, but its expression level decreased significantly upon infection (Figure 6B); as such, this gene which was speculated to be a negative regulator of resistance to SCLB. *Zm00001d039531* and *Zm00001d018203* of the WRKY TF family appeared to be negatively regulated genes in both resistant and susceptible lines, because their expression levels were reduced after inoculation with the pathogen (Figure 6D).

### 2.8. Verification of DEGs by Quantitative RT-PCR (qRT-PCR)

qRT-PCR was performed to verify the expression profiles of DEGs from our RNA-seq data. Eight DEGs were randomly selected (Figure 7). The results showed the expression patterns of these genes at Mock, 1 dpi, and 3 dpi. Except for *Zm00001d028816* (pathogenesis-related protein 6) and *Zm00001d024843* (1-aminocyclopropane-1-carboxylate oxidase15), the other six genes were activated after inoculation with pathogens in susceptible or resistant lines. Four out of six genes, i.e., *Zm00001d048913* (calcium-binding protein CML45), *Zm00001d014649* (disease resistance protein), *Zm00001d007329* (WRKY-transcription factor 108), and *Zm00001d018734* (pathogenesis-related protein 1), were rapidly activated at 1 dpi, but at 3 dpi, their expression levels were reduced. *Zm00001d028816* was more important in the susceptible line SK39, and its expression level promptly increased when activated by pathogens, especially at 3 dpi. The expression of *Zm00001d024843* in the resistant line RK13 was always kept at a relatively low level, whereas in the susceptible line, SK39 increased first and then decreased. The expression patterns of qRT-PCR were generally consistent with those of RNA-seq, demonstrating the accuracy of RNA-seq analysis.

## 3. Discussion

Southern corn leaf blight (SCLB) damages maize leaf tissue. The lesions are related to PPO [10]. We observed that the onset time was earlier in the susceptible line SK39. At 1 dpi, larger lesions appeared in the middle of the leaves of the susceptible line SK39; at 3 dpi, the lesions showed a spreading trend, and the leaf tips appeared dry and necrotic (Figure 1A). The resistant line RK13 demonstrated less severe disease symptoms: at 3 dpi, the leaves turned yellow and numerous large spots appeared (Figure 1B). Our results were largely consistent with the pathogenic characteristics of resistant and susceptible lines. We also observed that PAL and PPO activities were not associated with cultivar resistance. This is consistent with the results of previous studies [4]. Vanitha et al. [12] observed changes in PAL and PPO activity after vaccination, but no significant correlations were observed between these trends and resistance traits.

SCLB is a typical foliar disease, and we observed photosynthesis in its disease resistance process (Figure 3 and Figure 4). Two photosynthesis-related terms (“photosynthesis, light harvesting” and “photosynthesis, light reaction”) and two defense-related terms (“defense response” and “response to biotic stimulus”) were also significantly enriched (Figure 4B). Photosynthesis and immune defense are interconnected in complex networks [27]. Previous studies have demonstrated that photosynthesis is the main pathway in plant–pathogen interaction processes [28,29]. From a phytopathological perspective, changes in photosynthesis in the early stages of pathogen infection may be a causal factor affecting acquired resistance, while those in later stages may be the result of resistance formation [30]. Photosynthesis generates ATP, NADPH, and carbohydrates that can be used to synthesize defense-related hormones such as abscisic acid (ABA), ethylene (ETH), jasmonic acid (JA), and salicylic acid (SA), as well as antimicrobial compounds [31]. In addition, the chloroplast is a major producer of JA and SA, and is also the site of Ca^2+^ signaling (Figure 5A). These signaling molecules are also essential for plant defense [32], and we observed that JAZ, CaM4, and PR were activated upon inoculation with pathogens (Figure 5B). Moreover, “Gene silencing” was significantly enriched in our study (Figure 4B), leading us to speculate that ABA may be involved in disease resistance via RNA silencing during pathogen invasion [33].

Mitogen-activated protein kinase (MAPK) pathways transfer biotic stress information from sensors to cellular responses [34]. We found that the MAPK signaling pathway was enriched in both lines, but was more significant in the resistant line (Figure 3B). The immune signaling MAPK pathway forms with *MAPK3*/*MAPK5*, *MKK4*/*MKK5*, and *MPK3*/*MPK6* have been reported to transduce defense signaling downstream of multiple plant receptor kinases [35]. The MAPK signaling pathway may only play a role in transmitting pathogen invasion signals to target cells, but disruption of this transmission process can also lead to reduced plant resistance to pathogens. A good illustration is that the *GhMKK6*-*GhMPK4* cascade signaling pathway plays an important role in cotton resistance to fusarium wilt by regulating SA or JA-mediated defense pathways. Silencing *GhMPK4* reduced cotton tolerance to fusarium wilt and decreased the expression of several resistance genes [36,37,38].

SA, a phenylpropane-derived compound, plays a crucial role in plant-disease resistance and can act as an inducer, enabling plant endogenous signal-like molecules to trigger the expression of pathogenesis-related (PR) genes (Figure 5A), which are responsible for the production of hypersensitive reactive (HR) systemic acquired resistance (SAR) in plants [39,40]. We found that the expression levels of PR1 protein genes greatly increased following inoculation with pathogens (Figure 5B). Furthermore, PAL, a key enzyme in the phenylpropane biosynthetic pathway (Figure 5A), enhances SA-triggered SAR. Overexpression of *NbPAL* in tobacco has been reported to significantly reduce the susceptibility of plants to *Cercospora nicotianae* [41]. *CaPAL1* in pepper was reported as a positive regulator of SA-dependent defense signaling against microbial pathogens via its enzymatic activity in the phenylpropane pathway [42]. Furthermore, SA can increase the activities of plant protective enzymes such as SOD and POD, thereby reducing plant damage by ROS (Figure 5A).

*C. heterostrophus* is a necrotrophic pathogen, and its invasion results in necrosis in mesophyll cells (Figure 1A,B). The activation of the JA signaling pathway is required for resistance against necrotrophic pathogens. ZIM genes have been implicated in JA responses; for example, JASMONATE ZIM-domain (JAZ) proteins act as transcriptional repressors of JA responses and play a crucial role in the regulation of host immunity in plants [43,44]. Our study found that most JAZ genes were highly expressed in resistant lines RK13 (Figure 5B), which was consistent with the expression pattern of ZIM TFs (Figure 6E). The role of JAZ is twofold: *JAZ1* was identified as a favorable gene for enhancing resistance to powdery mildew through promoting ROS accumulation in bread wheat [45], whereas overexpression of *JAZ8* suppressed plant defense responses against *B. cinerea* [46]. In addition, *JAZ8* represses the transcriptional function of *WRKY75*, thereby reducing plant disease resistance. *WRKY75*, as an important component of JA-mediated signaling pathway, positively regulates *Arabidopsis* defense responses against necrotizing pathogens [47]. Consistently, the WRKY TFs were mainly highly expressed in the susceptible line SK39 (Figure 6). A striking term, “response to oxidative stress”, indicates that diseased plants take active measures to mitigate ROS damage (Figure 4B). We conjecture that the resistant line RK13 may increase the accumulation of ROS via enhancing the transcription of JAZ, thereby improving resistance. In the susceptible line SK39, the level of WRKY was elevated by inhibiting the expression of JAZ, which enhanced defense responses against the pathogen.

To overcome the biotic stress caused by pathogenic infection, plants produce different kinds of secondary metabolites (phenylpropanes, quinones, flavonoids, and terpenoids) [46,48] (Figure 5A). Metabolomic analyses have provided some insights into substances that are resistant to SCLB. A characterization of metabolites in maize lines with different SCLB resistance by ^1^H-NMR-FTIR spectroscopy showed that polyphenols, flavonoids, and lignin might be responsible for the induction of resistance to SCLB, and threonine, γ-amino butyric acid, malic acid, and fatty acids could be considered as potential biochemical markers of SCLB susceptibility [49]. A negative correlation was found between terpenoids and fungal infestation, and it was found that SCLB (*C. heterostrophus*) strongly elicited zealexin A4 (ZA_4_) and kauralexin diacids [50]. Studies have shown that hormone signaling molecules such as SA and JA are not only involved in regulating the biosynthesis of plant secondary metabolites (Figure 5A), but also interact with them to form a complex signaling network [51,52].

Transcription factors (TFs) play important roles in plant defense response pathways. We identified 230 differentially expressed TFs from the 2499 co-expressed DEGs (Appendix A). The bHLH, MYB, NAC, WRKY, and ZIM families make up the largest number of TF families (Figure 6). The top four are the most reported TF families related to plant disease resistance. Notably, WRKY is the TF family which has been most frequently reported to activate to plant disease resistance; its members are involved in the regulation of transcriptional reprogramming of plant immune responses [53,54]. Transactivation mediated by the MAPK-WRKY pathway has been reported to be required for pattern-triggered immunity (PTI) and effector-triggered immunity (ETI) [55]. We found that the expression of MYB TFs in susceptible line SK39 was very significant (Figure 6B), which may play an important role in its susceptibility. *CsMYB96* has been reported to enhance fungal pathogen resistance in citrus and arabidopsis by promoting SA biosynthesis and the accumulation of defense metabolites [56]. *TaPIMP2* encodes a wheat pathogen-induced MYB protein that positively regulates the expression of pathogenesis-related genes, such as *PR1*, *PR2*, *PR5*, and *PR10* (important elements in the SA signaling pathway) and enhances plant defense responses to pathogens [57]. Thus, differentially expressed MYB genes may be involved in the SA signaling pathway by regulating the expression of PR genes and the biosynthesis of defense metabolites in susceptible lines. Furthermore, we discovered that the expression patterns of ethylene- and auxin-related TFs differed between the resistant and susceptible lines. Ethylene transcription factors mainly play a role in susceptible lines, whereas auxin transcription factors act on the resistant lines (Figure 6F).

## 4. Materials and Method

### 4.1. Plant Materials

In this study, sweet corn inbred lines, i.e., resistant K13 (RK13) and susceptible K39 (SK39), were planted in the Experimental Teaching Base of South China Agricultural University (Zengcheng, Guangdong, China; 113.81° N, 23.13° E) in autumn, 2020. Both varieties were provided by the laboratory of sweet corn genetic improvement (College of Agriculture, South China Agricultural University, Guangzhou, Guangdong, China). Treated RK13 and SK39 seeds (soaked in 7% sodium hypochlorite solution for 30 min and rinsed three times with sterile water) were sown in sterilized pots containing nutrient soil and then cultured in an artificial climate chamber (25 °C, 75% relative humidity, 16 h light/8 h dark).

Four-leaf stage RK13 and SK39 plants were inoculated with *C. heterostrophus* as previously described [58], and infected leaves were collected without pathogens (Mock) and at 1- and 3-days post inoculation (1 dpi and 3 dpi). The strain used in the experiment was the physiological race O of the highly virulent SCLB, which was provided by Professor Canwei Shu (College of Plant Protection, South China Agricultural University, Guangzhou, Guangdong, China). Mosaic leaves were observed when the leaves were infected with pathogens. For each sample, leaves from at least three separate plants were collected with three biological replicates. 

### 4.2. Determination of Plant Protective Enzyme Activity

SOD (superoxide dismutase) enzymatic activity was measured using an improved WST method. The reaction solution had characteristic light absorption at 450 nm, and the color depth was inversely proportional to SOD activity (the darker the color, the lower the SOD activity). POD (peroxidase) catalyzes the oxidation reaction of H_2_O_2_, and the reaction produces a reddish-brown product, which has a characteristic light absorption at 470 nm; POD activity can be determined according to this characteristic. Polyphenol oxidase (PPO) is a copper-containing oxidase, and the reaction product after catechol catalysis has a characteristic light absorption at 420 nm, which can be used to determine the activity of PPO. PAL (phenylalanine ammonia lyase) catalyzes the cleavage of L-phenylalanine. The product of this reaction has characteristic light absorption at 290 nm, and the activity of PPO can be determined according to this characteristic. The kits for measuring enzymes activity were provided by Grace Biotechnology (Suzhou, Jiangsu, China), and the assays were performed according to the manufacturer’s instructions. Three biological replicates were created for each set of experiments.

### 4.3. RNA Extraction and Library Preparation

The total RNA was extracted from the samples using a Plant Total RNA Purification Kit (RC401-01, Vazyme, Nanjing, Jiangsu, China) following the manufacturer’s protocols. To ensure the samples can be used for transcriptome sequencing, the Nanodrop 2000 spectrophotometer (Thermofisher, Waltham, MA, USA), Qubit 2.0 Fluorometer (Life Technologies, Carlsbad, CA, USA), and Agilent 2100 Bioanalyzer system (Agilent Technologies, Santa Clara, CA, USA) were used to evaluate the purity (OD260/280 ≥ 1.8; OD260/230 ≥ 1.0), concentration (total RNA ≥ 250 ng/µL), and integrity (RIN ≥ 8.0, 28S/18S ≥ 1.5) of RNA samples. DNA library generation and RNA-seq high-throughput sequencing were performed by Majorbio Biotech (Shanghai, China). A total of 18 qualified libraries were sequenced on the Illumina Novaseq platform HiSeq^TM^ 2500 (Illumina, San Diego, CA, USA), and 150 bp paired-end reads were generated.

### 4.4. Preprocessing of RNA-Seq Data

Raw sequence reads were processed using the FastQC package (https://www.bioinformatics.babraham.ac.uk/projects/fastqc, accessed on 25 June 2022) [59] to remove reads containing poly-N, low-quality or adaptor-polluted reads. The clean reads were mapped to maize B73 reference genome (version 4.0) using Cufflinks (http://cole-trapnell-lab.github.io/cufflinks/, accessed on 25 June 2022) [60]. The number of fragments per kilobase of transcript per million mapped reads (FPKM) value for each gene was calculated using RSEM (http://deweylab.github.io/RSEM/, accessed on 25 June 2022) [61].

### 4.5. Identification of DEGs and Functional Classification

Differentially expressed Genes (DEGs) between two samples were identified using R package DESeq2 [62]. The threshold for defining DEGs was as follows: adjusted *p*-value < 0.05 and fold change > 2; only those genes with an FPKM value ≥ 1 in at least one stage were retained for further analysis. To identify the putative biological functions and pathways of DEGs, Gene Ontology (GO) annotation (http://geneontology.org/, accessed on 25 June 2022) [63] and Kyoto Encyclopedia of Genes and Genomes (KEGG) pathway analysis (https://www.kegg.jp/, accessed on 25 June 2022) [64] were performed. Time series analysis of DEGs using the STEM (Version 1.3.11, http://www.cs.cmu.edu/~jernst/stem/, accessed on 27 June 2022). All these analyses were carried out using the online OmicShare tools (GENE DENOVO, Guangzhou, China, http://www.omicshare.com/tools, accessed on 28 June 2022).

### 4.6. Real-Time PCR Analysis

Total RNA was extracted from the leaves at 0 dpi, 1 dpi, and 3 dpi. The first strand of cDNA was synthesized according to the instructions for the Goldenstar^TM^ RT6 cDNA synthesis Mix (TIANGEN, Beijing, China). Subsequently, qRT-PCR was performed using the SYBR PrimeScript RT-PCR Kit (TIANGEN, Beijing, China) with SYBR Green dye. The maize actin gene ZmActin was used as the internal control. The 2^−ΔΔCT^ [65] quantitative analysis method was used to calculate the relative expression level. The primers used in this study are listed in Appendix A.

## 5. Conclusions

In the present study, transcriptome sequencing of southern leaf blight in sweet corn at different stages of infection was performed. A functional analysis of DEGs provided a dynamic view of transcriptional variation in sweet corn against southern leaf blight infection. Five pathways, i.e., “photosynthesis”, “plant hormone signal transduction”, “MAPK signaling pathway”, “phenylpropanoid biosynthesis”, and “biosynthesis of secondary metabolites”, as well as transcription factor families, were shown to play a crucial role in disease resistance. These results provide a valuable reference for understanding the disease resistance mechanisms of sweet corn during southern leaf blight infection and for determining disease resistance genes. Further research will be necessary to understand the functions of DEGs and pathways during southern leaf blight response to unravel the mechanism of resistance in maize.

## Figures and Tables

**Figure 1 ijms-23-10236-f001:**
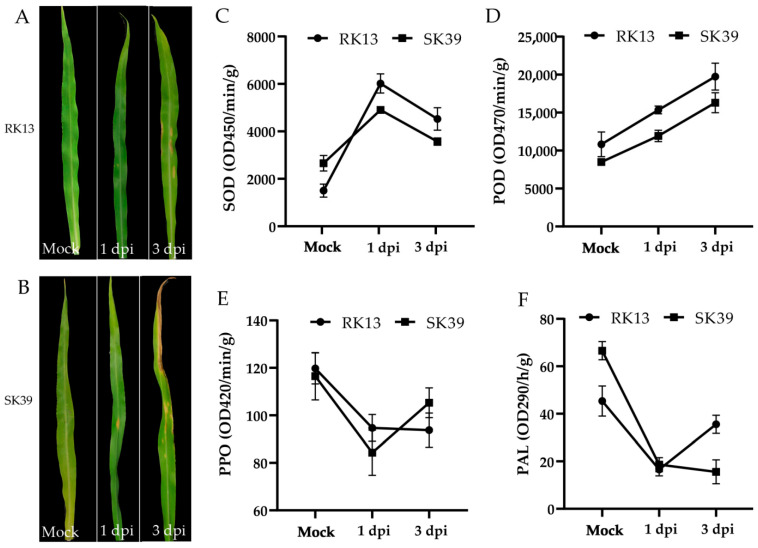
Morphological characteristics and changes in protective enzymes of leaves without pathogens (Mock) and with *C. heterostrophus* after inoculation (1 dpi, 3 dpi). (**A**,**B**) Morphological characteristics of leaves in RK13 (**A**) and SK39 (**B**). (**C**–**F**) The changes in protective enzymes (SOD, POD, PPO, and PAL) after inoculation with pathogens.

**Figure 2 ijms-23-10236-f002:**
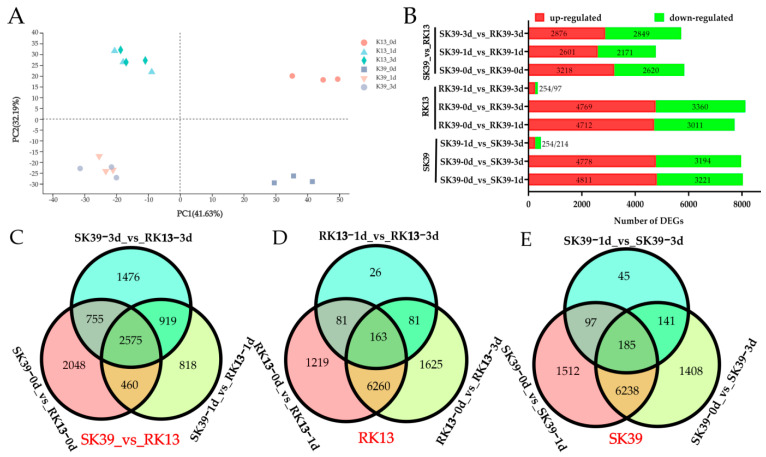
Comparative analysis of differentially expressed genes (DEGs) in the sweet corn response to southern corn leaf blight. (**A**) Principal component analysis of the transcriptome data in RK13 and SK39 treated with the *C. heterostrophus*. (**B**) The numbers of up- and down-regulated DEGs in the different comparison groups. (**C**) Venn diagram of DEGs in the SK39_vs_RK13 group (includes three pairwise combinations, SK39−0d_vs_RK13−0d, SK39−1d_vs_RK13−1d, and SK39−3d_vs_RK13−3d). (**D**) Venn diagram of DEGs in the RK13 group (includes three pairwise combinations, RK13−0d_vs_RK13−1d, RK13−1d_vs_RK13−3d and RK13−1d_vs_RK13−3d). (**E**) Venn diagram of DEGs in the SK13 group includes three pairwise combinations, (SK39−0d_vs_SK39−1d, SK39−1d_vs_SK39−3d and SK39−1d_vs_SK39−3d). Red labels represent group names.

**Figure 3 ijms-23-10236-f003:**
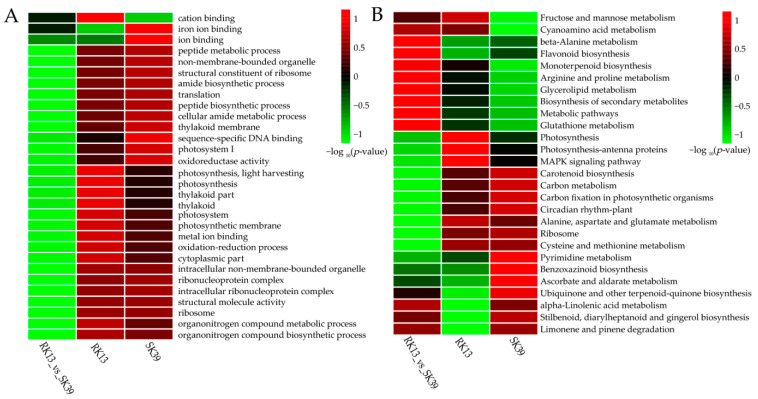
Functional classification of DEGs. Heatmaps of the Top 30 GO terms (**A**) and the Top 30 KEGG pathways (**B**) that were enriched in two or more comparison groups. The darker the red, the more significant; green indicates low significance.

**Figure 4 ijms-23-10236-f004:**
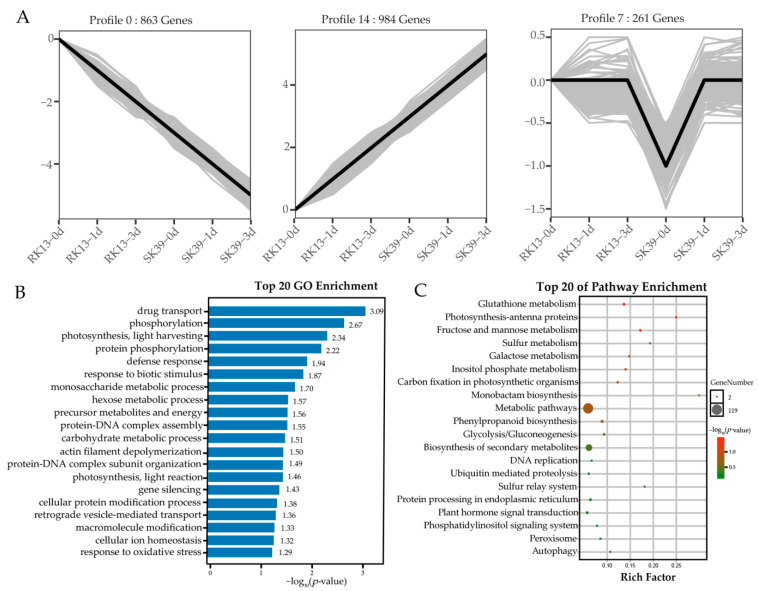
Clustering analysis of co-expressed DEGs in the three groups (RK13, SK39 and SK39_vs_RK13). (**A**) Gene expression patterns across time points. Significant trends (*p* < 0.05) were presented. The gray and black lines represent each gene’s expression pattern and the expression trend of all genes, respectively. (**B**) GO enrichment analysis of co-expressed DEGs. Blue bars represent the significance of terms (*p*-value). (**C**) The KEGG pathway enrichment of co-expressed DEGs. The size of the circle represents the number of genes enriched. The color of the circle represents the significance of the pathway; the darker the red the more significant, while green means low significance.

**Figure 5 ijms-23-10236-f005:**
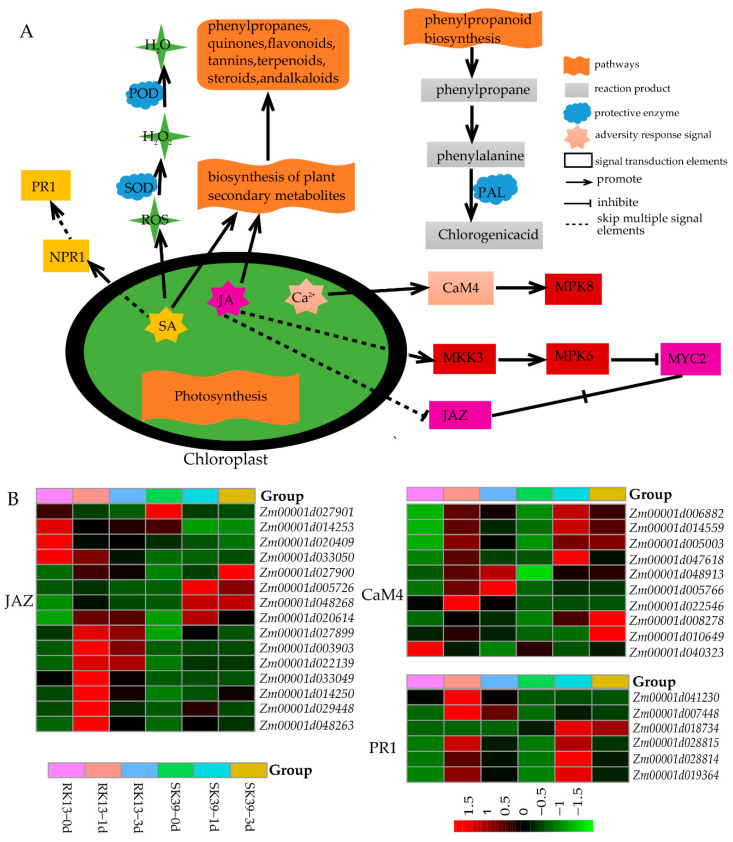
The important pathways of sweet corn response to SCLB and the gene expression patterns of related elements. (**A**) The important pathways of sweet corn response to SCLB. (**B**) The gene expression patterns of related elements. Red means high expression, green means low expression.

**Figure 6 ijms-23-10236-f006:**
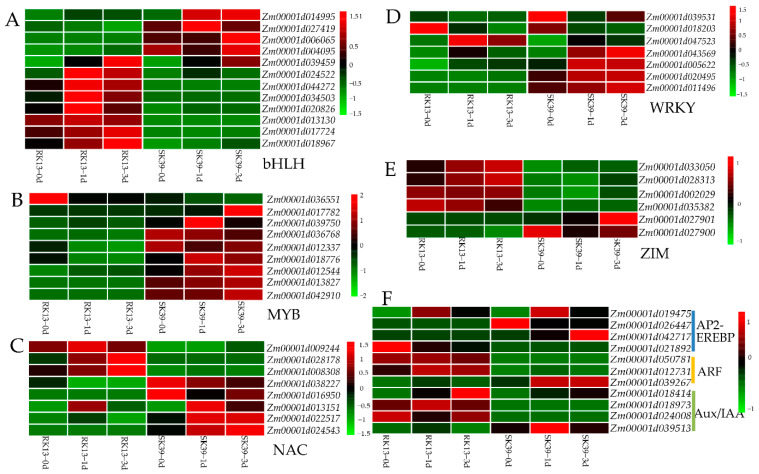
Heatmap of differentially expressed TFs. Red means high expression, green means low expression. (**A**) Gene expression patterns of bHLH TF numbers. (**B**) Gene expression patterns of MYB TF numbers. (**C**) Gene expression patterns of NAC TF numbers. (**D**) Gene expression patterns of WRKY TF numbers. (**E**) Gene expression patterns of ZIM TF numbers. (**F**) Gene expression patterns of pant hormone response TF families.

**Figure 7 ijms-23-10236-f007:**
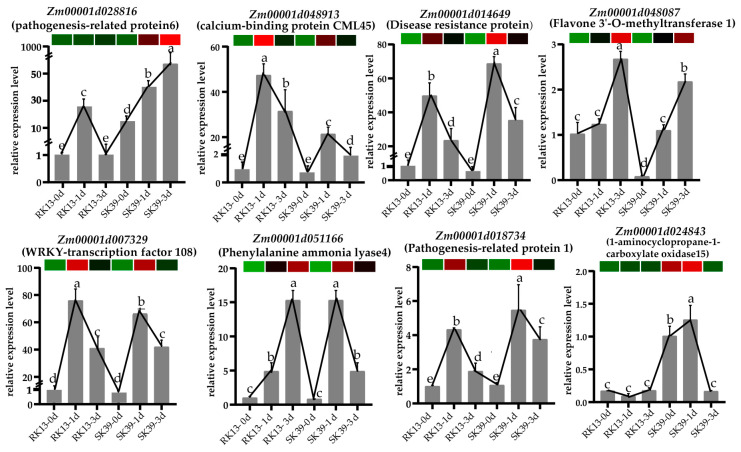
qRT-PCR validation of DEGs at different time points (0 dpi, 1 dpi and 3 dpi). Heatmaps are based on FPKM of transcriptome data. Lines represented gene expression changes in different samples. Values followed by different lowercase letters are significantly different at *p* < 0.05, and the error line is the standard error.

**Table 1 ijms-23-10236-t001:** Summary of mRNA sequencing datasets.

Sample	RK13_0d	RK13_1d	RK13_3d	SK39_0d	SK39_1d	SK39_3d
Raw reads (10^6^)	45.53	45.06	46.38	46.13	46.47	45.87
Clean reads (10^6^)	45.30	44.84	46.17	45.88	46.24	45.65
Error rate (%)	0.0246	0.0246	0.0245	0.0245	0.0248	0.0247
Total mapped (%)	90.22	88.56	86.21	90.28	89.13	87.56
Multiple mapped (%)	3.57	2.93	3.69	3.73	3.39	3.04
Unique mapped (%)	86.65	85.63	82.52	86.55	85.75	84.53
Q30 (%)	94.56	94.56	94.64	94.65	94.37	94.43
GC content (%)	56.97	55.49	55.54	56.20	55.78	55.69

## Data Availability

All the transcriptome sequencing data of this study are available in the National Center for Biotechnology Information Sequence Read Archive database with the accession number PRJNA853857.

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
