# Peer review of "Physiological and Molecular Characteristics of Southern Leaf Blight Resistance in Sweet Corn Inbred Lines"

_ijms, 2022, doi:10.3390/ijms231810236_

Round 1
Reviewer 1 Report
The manuscript is clearly written, only a few things need to be fixed:
4.1. Please list the GPS for the locations mentioned in this paragraph.
What is the origin of C. heterostrophus?
General question: do the authors believe that prokaryotes on the leaves can have any effect on your observations - on the fungi infection etc?
Author Response
Dear Reviewer,
We would like to thank you for the letter of correspondence dated August 16, 2022, and the opportunity to resubmit a revised manuscript (ijms-1841107). We would also like to express our gratitude to you for your comments that have surely contributed to a more complete manuscript.
We have carefully considered each comment, recommendation, and suggestion and have made the following key revisions to the manuscript:
- We have reorganized the abstract section to make it clearer.
- We deleted most of the discussion and have rewritten it to make it more relevant to the study.
- The results and figure legends were modified to improve the accuracy of the descriptions based on this suggestion.
- More detailed information has been provided, including about DEGs and the comparison samples.
Please find below our point-by-point responses to your comments.
We believe that these revisions have improved the paper and hope that you and the reviewers will consider it for publication in Int. J. Mol. Sci. The manuscript has benefited greatly from your insights. Thank you for the consideration and effort in helping us improve our paper.
Sincerely yours,
Dr. Jun Huang
Jun Huang and colleagues (junhuang@scau.edu.cn)
The Key Laboratory of Plant Molecular Breeding of Guangdong Province, South China Agricultural University, No.483, Wushan Street, Tianhe District, Guangzhou 510642, P.R. China
Q1.4.1. Please list the GPS for the locations mentioned in this paragraph.
Response: Thank you for your suggestion. The GPS for the locations mentioned in 4.1 has been added in this revised version as follows:
In this study, sweet corn inbred lines, resistant K13 (RK13) and susceptible K39 (SK39) were planted in the Experimental Teaching Base of South China Agricultural University (Zengcheng, Guangdong, China; 113.81°N, 23.13°E) in autumn, 2020.
Q2.What is the origin of C. heterostrophus?
Response:
The strain used in the experiment was the physiological race O of the highly virulent SCLB, which was provided by Professor Canwei Shu (College of Plant Protection, South China Agricultural University, Guangzhou, Guangdong, China)
Q3.General question: do the authors believe that prokaryotes on the leaves can have any effect on your observations - on the fungi infection etc?
Response: You raise an interesting point about whether prokaryotes effect on fungi infection on the leaves. Cochliobolus heterostrophus is a fungal plant pathogen which cause southern corn leaf blight in maize. Fungi and prokaryotes, such as bacteria, are found living together in a wide variety of environments. Number studies reported that the interaction between fungi and prokaryotes, such as bacteria are significant drivers of many ecosystem functions and are important for the health of plants. In my opinion, prokaryotes might be having some effect during C. heterostrophus infection to maize (I am not sure, no related papers were found). However, the resistant K13 (RK13) and susceptible K39 (SK39) identified in the present study were in the same growth condition. Therefore, even the interaction between C. heterostrophus and prokaryotes was occurred, in my opinion, it does not affect on the observation. Thank you for your valuable comment.

Author Response
Dear reviewers and editor,
We would like to thank you for the letter of correspondence dated August 16, 2022, and the opportunity to resubmit a revised manuscript (ijms-1841107). We would also like to express our gratitude to you for your comments that have surely contributed to a more complete manuscript.
We have carefully considered each comment, recommendation, and suggestion and have made the following key revisions to the manuscript:
- We have reorganized the abstract section to make it clearer.
- We deleted most of the discussion and have rewritten it to make it more relevant to the study.
- The results and figure legends were modified to improve the accuracy of the descriptions based on this suggestion.
- More detailed information has been provided, including about DEGs and the comparison samples.
Please find below our point-by-point responses to your comments.
We believe that these revisions have improved the paper and hope that you and the reviewers will consider it for publication in Int. J. Mol. Sci. The manuscript has benefited greatly from your insights. Thank you for the consideration and effort in helping us improve our paper.
Sincerely yours,
Dr. Jun Huang
Jun Huang and colleagues (junhuang@scau.edu.cn)
The Key Laboratory of Plant Molecular Breeding of Guangdong Province, South China Agricultural University, No.483, Wushan Street, Tianhe District, Guangzhou 510642, P.R. China
Point by Point Response
The article titled:
"Physiological and Molecular Characteristics of Southern Leaf Blight Resistance in Sweet Corn Inbred lines":
demonstrated a promising molecular mechanism of southern leaf blight resistance in sweet corn through an RNA-seq based strategy.
The experimental data showed that “plant hormone signal transduction”, “MAPK signaling pathway”, etc, is involved in the southern leaf blight resistance. They verified some important genes to prove the hypothesis that the genes might play a key role in the resistance formation.
I think the article was well written and the data was sufficient to support the title, and the results help to dissect the real mechanism of southern leaf blight resistance in sweet corn or in other corps.
As we know, transcription data is insufficient to reveal the mechanism of biological process, much more work should be focused on the functional studies to key genes or proteins.
I hope the authors continue the research based on the present findings to investigate some useful points.
The article measured some enzymes’ activities, including POD, SOD. Obviously, the authors wanted to show that both enzymes are involved in the process. However, more details should be addressed to relationship between the different expression of these two enzymes and the different resistance of two genotypes.
Needs to add some more recent literature in introduction and discussion sections, please:
Puttarach J, P. Puddhanon, S. Siripin, V. Sangtong and S. Songchantuek (2016). Marker assisted selection for resistance to northern corn leaf blight in sweet corn. SABRAO J. Breed. Genet. 48(1): 72-79.
Al-Tamimi AJT (2020). Genetic variation among Zea mays genotypes using start codon targeted marker polymorphism. SABRAO J Breed Genet 52(1): 1-16.
Mushtaq M, Bhat MA, Bhat, JA, Mukhtar S, Shah AA. (2016). Comparative analysis of genetic diversity of maize inbred lines from Kashmir Valley using agro-morphological and SSR markers. SABRAO J. Breed. Genet. 48(4): 518–527.
Puttarach J, P. Puddhanon, S. Siripin, V. Sangtong and S. Songchantuek (2016). Marker assisted selection for resistance to northern corn leaf blight in sweet corn. SABRAO J. Breed. Genet. 48(1): 72-79.
Li, H., N. Gou, C. Chen, K. Liu, L. Wang and T. Wuyun. 2022. Genome-wide identification and characterization of the Dof gene family in Prunus sibirica. Pak. J. Bot., 54(4): 1375-1383. DOI:http://dx.doi.org/10.30848/PJB2022-4(33)
Shahrasbi, S., H.P. Anosheh, Y. Emam, M. Ozturk and V. Altay. 2021. Elucidating some physiological mechanisms of salt tolerance in Brassica napus L. seedlings induced by seed priming with plant growth regulators. Pak. J. Bot., 53(2): 367-377. DOI: http://dx.doi.org/10.30848/PJB2021-2(34)
Liping, R., Y. Dandan, W. Wenyang, F. Tingting, S. Xiaohui and C. Xiaohan. 2022. Cloning, characterization, and expression analysis of two MAPKKK genes in Chrysanthemum. Pak. J. Bot., 54(1): 143-150, DOI:http://dx.doi.org/10.30848/PJB2022-1(3)
Over all paper is good and worth for publication.
Response: Thank you for taking the time and energy to help us improve the paper. We carefully considered your comments as well as those offered by another reviewer. In our revision, based on the comments and recommendations, we have incorporated the following changes:
- We have reorganized the abstract section to make it clearer.
- We deleted most of the discussion and have rewritten it to make it more relevant to the study.
- The results and figure legends were modified to improve the accuracy of the descriptions based on this suggestion.
- More detailed information has been provided, including about DEGs and the comparison samples.
Moreover, I agree that the present transcription data is insufficient to reveal the mechanism of biological process, further research will be necessary to understand the functions of key DEGs to unravel the mechanism of resistance to southern leaf blight resistance in sweet corn. We also emphasize it in the Discussion section. Thank you for your suggestion.
We also carefully read the literature you mentioned and gained a lot. We have quoted several papers in the revised version. Such as: Marker assisted selection is still an important method for mining plant disease resistance genes [13], and is widely used in the mapping of disease resistance genes of northern corn leaf blight (NCLB) [14] and southern corn leaf blight (SCLB). Mitogen-activated protein kinase (MAPK) pathways transfer biotic stress information from sensors to cellular responses [36]. Please check the revised manuscript.
Again, we appreciate all of your insightful comments.

Reviewer 3 Report
The authors have done considerable work in a difficult task. Identifying genes are involved in resistance and not just changed as result of resistance is a remarkably difficult task. They have compared the gene response of one susceptible maize line with one maize line that is more resistant. They found many differences. The differences may or may not be involved in resistance. They may have found the same differences if they compared two susceptible lines or two resistant lines. Different genotypes will have different responses. This study could possibly provide a first step in identifying genes involved in resistance, but it certainly does not characterize SCLB resistance in maize. I have empathy with the authors in attempting to understand the very complex reactions of plants to pathogen infection. Unfortunately, they have not achieved the goals identified in the abstract and in the title.
There are some confusions in the manuscript.
There are too many sentences in the discussion that are devoted to generalities that are not directly related to the study. These sentences are more appropriate to a textbook on disease resistance.
L22-26. These groups need more description. Without more information, a reader cannot know what they are.
L29. Change “resistance” to “response”. The authors have not demonstrated that these pathways are involved in resistance. Perhaps they are, but this is not demonstrated. Other pathways (i.e. drug transport) were also demonstrated to change in these groups. Why were these other pathways ignored?
L35-48. The authors need to clarify that the subject of this study is not resistance to the toxin, but rather to the pathogen in maize not containing T-cytoplasm.
L51. Replace “is” with “are”.
L88-89. Replace "response” with “responds”.
L98-110. Identify explicitly that RK13 is the resistant and SK39 is the susceptible line.
L105-110. This section is confusing because long sentences are used and disparate thoughts are included in a single sentence. How many times was this experiment done? Are the graphs the result of a single experiment? POD increased with time. SOD increased then abated. PPO decreased and PAL decreased.
L119. What is meant by “pre-inoculation (wild-type, WT)”?
L132. From the Methods section, it is not clear to me that in fact these are biological replicates. They appear to be technical replicates. The experiment has to be done three separate times to have three biological replicates.
L148. I question whether all of the genes upregulated are involved in resistance. They might be disease responsive, but the authors have not demonstrated involvement in resistance.
L159. There is no evidence that the differences are due to resistance. The authors have compared to different maize genotypes that differ in many ways – not just in “resistance”. There would be many differences between two resistant lines of different genotypes. Remove this assertion of resistance.
L159-161. The labeling of Figure 2 is confusing. Don’t the authors mean Figure 2 C, D, &E? The colors in these figures need to be explained.
L195. What is “susceptibility time”?
L202-204. The top three GO terms were related to drugs. Why were these not mentioned?
L205-206. I don’t understand what was speculated or what “enhancing resistance from forward and reverse, respectively” means.
L252. Delete the “not”.
L274-276. The rationale for this statement needs to be provided.
L277. In Figure 4, what do the colors represent?
L283-292. This information is more appropriate for the Introduction.
L294-308. Most of this information is not needed or is more appropriate in the Introduction.
L313-330. Most of this paragraph is not needed and hot helpful. It is also the case that as cells die due to infection, photosynthesis is interrupted.
L331-344. Delete. I don’t see how these paragraphs relate to the results of the present study.
L345-410. Delete. These are generalities that are not related by the authors to their study.
Author Response
The authors have done considerable work in a difficult task. Identifying genes are involved in resistance and not just changed as result of resistance is a remarkably difficult task. They have compared the gene response of one susceptible maize line with one maize line that is more resistant. They found many differences. The differences may or may not be involved in resistance. They may have found the same differences if they compared two susceptible lines or two resistant lines. Different genotypes will have different responses. This study could possibly provide a first step in identifying genes involved in resistance, but it certainly does not characterize SCLB resistance in maize. I have empathy with the authors in attempting to understand the very complex reactions of plants to pathogen infection. Unfortunately, they have not achieved the goals identified in the abstract and in the title.
Response: Thank you for taking the time and energy to help us improve the paper. We carefully considered your comments as well as those offered by another reviewer. In our revision, based on the comments and recommendations, we have incorporated the following changes:
- We have reorganized the abstract section to make it clearer.
- We deleted most of the discussion and have rewritten it to make it more relevant to the study.
- The results and figure legends were modified to improve the accuracy of the descriptions based on this suggestion.
- More detailed information has been provided, including about DEGs and the comparison samples.
As mentioned by reviewer 1#, we agree that the present transcription data are insufficient to identify specific genes that are involved in resistance because many differences exist between susceptible and resistant lines. Thus, we also compared gene expression levels between the samples and controls in both susceptible and resistant lines to explore the genes/pathways that were induced or depressed after inoculation. Moreover, we compared the transcriptome profiles of susceptible and resistant lines. This information provides a comprehensive overview of transcriptional regulation in the SCLB response in maize. As you mentioned, this study provides the first step in identifying the genes involved in resistance to SCLB. The data presented in the current study can also act as a resource for researchers interested in SCLB resistance in maize. We believe that the identification of key DEGs and pathways will guide future research. Further studies are necessary to understand the functions of key DEGs and pathways to unravel the mechanism of resistance to southern leaf blight resistance in sweet corn. Thank you for your valuable suggestions and comments.
There are some confusions in the manuscript. There are too many sentences in the discussion that are devoted to generalities that are not directly related to the study. These sentences are more appropriate to a textbook on disease resistance.
Response: We apologize for this confusion. Indeed, many sentences in the discussion are devoted to generalities that are not directly related to the study. In the revised manuscript, we have deleted most of the discussion and have rewritten this section to ensure that it is relevant to the study.
Q1. L22-26. These groups need more description. Without more information, a reader cannot know what they are.
Response: Thanks for your kind suggestion. We've added descriptions of these groups (“SK39” group, “RK13” group, and “SK39_vs_RK13” group).
Q2. L29. Change “resistance” to “response”. The authors have not demonstrated that these pathways are involved in resistance. Perhaps they are, but this is not demonstrated. Other pathways (i.e. drug transport) were also demonstrated to change in these groups. Why were these other pathways ignored?
Response: We agree your statement that we cannot demonstrate that these pathways are involved in resistance. We have modified it in the revision.
Q3. L35-48. The authors need to clarify that the subject of this study is not resistance to the toxin, but rather to the pathogen in maize not containing T-cytoplasm.
Response:
Response: Thanks for your kind suggestion. We have taken all your suggestions seriously and have made changes to the Introduction and Discussion of the manuscript. This content of the manuscript is not essential and may be misleading to readers, so we have removed it in the revision.
Q4. L51. Replace “is” with “are”.
Response: Thank you for your careful reading. We have modified it in the revised version.
Q5. L88-89. Replace "response” with “responds”
Response:We have modified it in the revised version.
Q6. L98-110. Identify explicitly that RK13 is the resistant and SK39 is the susceptible line.
Response: We have made it clear in the last paragraph of the Introduction and in the Result 2.1 that RK13 is the resistant and SK39 is the susceptible line.
Q7. L105-110. This section is confusing because long sentences are used and disparate thoughts are included in a single sentence. How many times was this experiment done? Are the graphs the result of a single experiment? POD increased with time. SOD increased then abated. PPO decreased and PAL decreased.
Response: Sorry for the confusion. The pathogens inoculation experiments were performed three times, and three technical replicates were set for each experiment. Changes in the activity of these protective enzymes are derived from this. Since there are few previous studies on the changes of protective enzyme activities during plant disease resistance, more references cannot be obtained, our experimental results are only representative of this study.
Q8. L119. What is meant by “pre-inoculation (wild-type, WT)”?
Response:“pre-inoculation (wild-type, WT)” means does not inoculate with pathogens. Sorry for the confusion. In the revision, we have modified this phrase to "mock" in the reversion..
Q9. L132. From the Methods section, it is not clear to me that in fact these are biological replicates. They appear to be technical replicates. The experiment has to be done three separate times to have three biological replicates.
Response:Thanks for your kind suggestion. To detect the activity of plant protective enzymes, we set up three independent time inoculation experiments. For each inoculation experiment we set up three technical replicates.
Q10. L148. I question whether all of the genes upregulated are involved in resistance. They might be disease responsive, but the authors have not demonstrated involvement in resistance.
Response: I totally agree your question. Not all up-regulated genes are involved in resistance to the pathogen. The responsive genes and pathways identified in the present study act as a resource for researchers interested in plant responses to southern leaf blight resistance in sweet corn. In future work, we believe it is worthwhile to investigate the role of responsive genes and pathways in the regulation of southern leaf blight response. Thus, we have modified the Discussion section in the revision.
Q11. L159. There is no evidence that the differences are due to resistance. The authors have compared to different maize genotypes that differ in many ways-not just in “resistance”. There would be many differences between two resistant lines of different genotypes. Remove this assertion of resistance.
Response: Yes. We also agree that there is no evidence that the differences are due to resistance Thus, we also compared gene expression levels between the samples and controls in both susceptible and resistant lines to explore the genes/pathways that were induced or depressed after inoculation. Moreover, we compared the transcriptome profiles of susceptible and resistant lines. We have removed the assertion of resistance. Thank you for your valuable suggestions and comments.
Q12. L159-161. The labeling of Figure 2 is confusing. Don’t the authors mean Figure 2 C, D, &E? The colors in these figures need to be explained.
Response:Thanks for your kind suggestion. We have revised the legend to Figure 2 and described the meaning of Figure 2 C, D, &E.
Q13. L195. What is “susceptibility time”?
Response: Sorry for inaccurate description, we have changed it to "infection time".
Q14. L202-204. The top three GO terms were related to drugs. Why were these not mentioned?
Response: Thanks for your kind suggestion. After we reanalyzed the results in this part, we removed duplicate GO terms. The new results are shown in Figure 4B. We found two new terms, "response to biotic stimulus" and "response to oxidative stress", which may be more representative of disease resistance processes. As for "GO terms were related to drugs", previous studies have not reported that it is related to plant disease resistance. Therefore, we did not focus on it in this study, but we will take it into account when mining disease resistance genes in the future.
Q15. L205-206. I don’t understand what was speculated or what “enhancing resistance from forward and reverse, respectively” means.
Response: "forward" means to enhance the plant's ability to resist pathogens through the activation of "defense response" genes; "reverse" means to inhibit the expression of genes that are conducive to pathogen infection through "gene silencing", thereby reducing the damage of pathogens to plants. we have changed this sentence to "enhancing resistance from direct activation of resistance genes and inhibition of susceptible genes, respectively".
Q16. L252. Delete the “not”.
Response: We've carefully checked and corrected it.
Q17. L274-276. The rationale for this statement needs to be provided.
Response: the result is now described as "The expression of Zm00001d024843 in the resistant line RK13 was always kept at a relatively low level, whereas in the susceptible line SK39 increased first and then decreased".
Q18. L277. In Figure 4, what do the colors represent?
Response: We have revised the legend in Figure 4.
Q19. L283-292. This information is more appropriate for the Introduction.
Response: We carefully consider and incorporate your comments. This information has been placed in the Introduction.
Q20. L294-308. Most of this information is not needed or is more appropriate in the Introduction.
Response: We carefully consider and incorporate your comments. This information has been placed in the Introduction.
Q21. L313-330. Most of this paragraph is not needed and hot helpful. It is also the case that as cells die due to infection, photosynthesis is interrupted.
Response: Thanks for your kind suggestion. We have deleted most contents and made substantial revision to ensure that it is relevant to the study. Please checked in the revised manuscript.
Q22. L331-344. Delete. I don’t see how these paragraphs relate to the results of the present study.
Response: We have revised the Discussion section to improve the relevance of results to discussion.
Q23. L345-410. Delete. These are generalities that are not related by the authors to their study.
Response: According your suggestion, we have revised the Discussion to improve the relevance of results to discussion.
